# Extracellular Vesicle-Based Nucleic Acid Delivery: Current Advances and Future Perspectives in Cancer Therapeutic Strategies

**DOI:** 10.3390/pharmaceutics12100980

**Published:** 2020-10-16

**Authors:** Crescenzo Massaro, Giulia Sgueglia, Victoria Frattolillo, S. Rubina Baglio, Lucia Altucci, Carmela Dell’Aversana

**Affiliations:** 1Department of Precision Medicine, University of Campania “Luigi Vanvitelli”, Via De Crecchio 7, 80138 Naples, Italy; crescenzo.massaro@unicampania.it (C.M.); giulia.sgueglia@unicampania.it (G.S.); victoria.frattolillo@studenti.unicampania.it (V.F.); 2Department of Pathology, Cancer Center Amsterdam, Amsterdam UMC, Vrije Universiteit Amsterdam, de Boelelaan 1117, 1081HV Amsterdam, The Netherlands; s.baglio@amsterdamumc.nl; 3Institute of Experimental Endocrinology and Oncology “Gaetano Salvatore” (IEOS)-National Research Council (CNR), Via Sergio Pansini 5, 80131 Naples, Italy

**Keywords:** extracellular vesicles, nucleic acid delivery, gene therapy, cancer

## Abstract

Extracellular vesicles (EVs) are sophisticated and sensitive messengers released by cells to communicate with and influence distant and neighboring cells via selective transfer of bioactive content, including protein lipids and nucleic acids. EVs have therefore attracted broad interest as new and refined potential therapeutic systems in many diseases, including cancer, due to their low immunogenicity, non-toxicity, and elevated bioavailability. They might serve as safe and effective vehicles for the transport of therapeutic molecules to specific tissues and cells. In this review, we focus on EVs as a vehicle for gene therapy in cancer. We describe recent developments in EV engineering to achieve efficient intracellular delivery of cancer therapeutics and avoid off-target effects, to provide an overview of the potential applications of EV-mediated gene therapy and the most promising biomedical advances.

## 1. Introduction

Extracellular vesicles (EVs) are nanoscale particles released by producing cells in the extracellular environment to interact with and act on distant and neighboring cells. EVs can transfer an assorted cellular cargo including DNA, messenger RNA (mRNA), microRNA (miRNA), and proteins to target recipient cells in the micro- and macroenvironment. Content heterogeneity is dependent not only on origin cell type but also on cell state (healthy, stressed, transformed, in replication) and may be modified by chemical or physical agents [1]. EVs have attracted growing interest within the scientific community in recent years [1], as their functional content is able to regulate or impair human physiological processes. EVs are in fact found in multiple types of extracellular fluids such as blood [2], saliva [3], urine [4], amniotic fluid [5], cerebrospinal fluid [6], and breast milk [7]. The physiological involvement of EVs has been gradually elucidated in immune response [8], signal transduction [9], angiogenesis [10], and cell proliferation [11]. Furthermore, several reports described the role of EVs as pathological effectors in cancer, where they appear to mediate virtually all aspects of tumorigenesis, including angiogenesis, immune suppression, metastasis formation, and drug resistance [12,13,14]. A better understanding of the functional mechanisms of EVs prompted the scientific community to explore the advantages of using EVs in the clinic for diagnostic [15], therapeutic [16], and prognostic [17] purposes. Primarily, researchers hypothesized introducing a desired cargo (drugs, proteins, nucleic acids) [18,19] into EVs to exploit their promising potential in cancer therapy. EVs preserve their content from degradation by keeping the cargo in a soluble phase in aqueous cytoplasm [20], and also resist enzymes in biological fluids in order to shield the cargo until it reaches target cells [21]. In addition, human EVs from HEK293 cells showed no toxicity and only weak immunogenicity when implanted in mice [22], both systemically and in the liver (one of most frequent site of non-engineered EVs accumulation) [23]. Therefore, one of the greatest advantages in the therapeutic use of EVs derives from their low-grade immunogenicity; for instance, EVs derived from mesenchymal stem cells (MSCs) are known to have a very low risk of immunogenicity. This particular ability could derive from the characteristics of the cells that originate them (MSC), showing a low expression of MHCII and other costimulatory molecules [24]. The use of these vesicles has been extensively studied in regenerative and anti-inflammatory therapy through the use of animal models such as for traumatic and perinatal brain injury [25]. Moreover, the MSC-EVs are widely used as therapeutic agents for different immunological disorders [26] and their use has shown incredibly promising results in clinical trials on severe forms of graft-versus-host disease (GvHD; Grade II–IV) [27]; chronic kidney diseases [28]; and Type I Diabetes Mellitus (T1DM) [29]. Another example of how EVs are able to negatively regulate immunoresponse is given by immature dendritic cell (DC)-derived EVs devoid of immunostimulatory molecules (MHC-I and -II) [30] and carrying co-regulatory proteins that suppress immune reactions. Further, DCs treated with immunosuppressive drugs or specific cytokines release EVs that negatively regulate immune cells, as shown by Kim SH et al., who found that EVs from DCs exposed to IL-10 suppressed inflammation and immune response [31]. Due to their size, EVs are able to avoid mononuclear phagocytic cell-mediated removal [32]. This contributes to increasing the longevity of EVs [33] and enhancing cargo bioavailability, as demonstrated in a very recent study by Carobolante et al. investigating the utility of EVs in improving the poor oral bioavailability of many drugs [34]. Additionally, the use of autologous vesicles likely avoids adverse reactions [35] and increases uptake efficiency by receiving cells [36]. EV uptake is highly efficient due to membrane proteins such as tetraspanins [37], fibronectin [38], and proteoglycans [37], but can also be customized depending on target cells. Vascular cell adhesion molecule 1- and integrin α4-enriched EVs enhance vesicle docking and uptake through endothelial cells [39]. Because of their small size, EVs also may cross the blood–brain barrier in a bi-directional manner, either by transcytosis (endocytosis followed by multivesicular body formation and release on the other side) [40,41] or through junctions between endothelial cells [40]. This passage is safe for EV cargo, as suggested by Alvarez-Erviti et al., who observed that EV-mediated small interfering RNAs (siRNAs) remain active after crossing the blood–brain barrier [42].

EVs may thus represent an optimal and safe alternative approach in current gene therapy. Gene therapy involves reverting aberrant gene expression, inducing the expression of suicide genes, modulating immune response, and inhibiting angiogenesis [43] by introducing DNA fragments, miRNA, siRNA, and lncRNA in target cells by different means such as viral vectors (human immunodeficiency virus (HIV), adeno-associated viruses) or naked plasmid DNA. Initial limited clinical success led to the development of a wide spectrum of safer, yet not fully efficient, gene-targeting delivery systems. Attractive nanoscale approaches in cancer gene therapy include polymeric, inorganic, lipid nanoparticles, and hybrid nanoparticles [44]. However, major attempts have been made to design and evaluate nanomaterials and nanotechnology to overcome delivery-related limitations such as adverse side effects, bio-accessibility, enzymatic degradation, non-selectivity, and premature release of therapeutics [45].

In this review, we describe the potential use of EVs as a tool for therapeutic nucleic acid delivery to target cells, to improve treatment efficacy and avoid side effects. We discuss the current state of and recent advancements in EV engineering, and the adoption of EVs as therapeutic vectors in tumor-targeted therapy.

## 2. Extracellular Vesicles (EVs) as a Delivery System for Nucleic Acids

The innate and unique proprieties of EVs in cell–cell communication strongly encourage studies into their usage as a therapeutic delivery system. EV-mediated transport seems to be highly effective, allowing the release of a greater amount of therapeutics into the tumor [46], thereby ensuring a lower amount in blood and reducing side effects [46]. EVs have a multifunctional utility in delivering therapeutics: they can signal through different modalities, by activating receptors on the surface of cells, delivering their content inside the cytoplasm, to carry hydrophilic molecules in their interior, lipophilic molecules in their lipid bilayers, and amphiphilic molecules on their surface.

A wide range of methods and technologies have been optimized in EV engineering [47]. Several strategies to load therapeutics inside EVs and functionalize them to cell targeting have been developed. A simplified schematic representation of how engineered therapeutic-containing EVs act is shown in Figure 1. Increasing evidence points to the compelling advantages over EV-producing cells [48,49], including no risk of transformation and lower immunogenicity. MSC-derived EVs were recently described as a novel cell-free therapy in liver, kidney, cardiovascular, and neurological diseases [50,51,52,53,54,55,56]. In addition, immature DCs were described as a major source of EVs [57] and as innovative cell-free vaccines, reverting cancer immune escape [58,59,60].

### 2.1. EV Therapeutic Engineering

Major benefits can be achieved through effective loading of EVs with therapeutic products. EV loading can be performed by two different means: either directly on purified vesicles or by engineering the mother cell to allow the release of modified EVs.

#### 2.1.1. EV Loading Methods

Purified EVs can be used to direct cargo loading with therapeutics in several ways. One of the most commonly used techniques for loading nucleic acids is electroporation [42]. The limitations of this approach are: (i) the aggregation of small RNAs, which can be partially prevented by using acidic citrate, found to work better than EDTA [61]; (ii) the low loading efficiency [55], even more pronounced for nucleic acids larger than 1000bp [62]. Consequently, alternative methods have been attempted to improve the loading efficiency of the molecule of interest. Hydrophobically modified siRNAs could be loaded into EVs: a cholesterol moiety (triethylene glycol (TEG)) was conjugated to the 3′-end of the desired siRNA, which was then co-incubated with purified EVs. As expected, modified siRNAs were found on the surface of purified vesicles, but a large percentage was also found inside EVs. Furthermore, modified siRNAs reduced the expression of their targets in receiving cells, thus proving functionality [63].

An advanced system was subsequently developed to load large nucleic acids into EVs by fusing exosomes with liposomes, obtaining a hybrid with the load-bearing capacity of liposomes and the ability of EVs to interact with and enter recipient cells. For example, CRISPR-Cas9 expression vectors were encapsulated in EVs in MSCs by co-incubation of exosomes, liposomes, and the plasmid of interest for 12 h at 37 °C. The loaded hybrid exosomes were taken up by MSCs and negative expression regulation of the target gene (*Runx2*) was observed in receiving cells, indicating functional delivery. However, some toxicity was detected due to the liposome component, which was not observed with EVs [64].

#### 2.1.2. Parental Cell-Based Engineering

The low loading efficiency of the above-mentioned methods for some large molecules can be overcome by engineering donor cells to release modified EVs [65]. Although several attempts have been made to better understand the intracellular system of EV sorting and loading, further studies are still required. EVs are known to be enriched in small RNAs [66], but the sorting mechanisms that actively and selectively incorporate RNAs into EVs are yet to be elucidated. Annexin A2 was reported to be involved in miRNA loading inside EVs [67]; sumoylated hnRNAPA2B1 binds to sequence motifs enriched in EV-associated miRNAs, mediating the loading of bound RNAs into EVs [68]. However, a “zipcode” sequence, involved in mRNA loading, was found in the 3′ UTR of mRNAs enriched in EVs [69]. Interestingly, miRNAs seem to play a role in mRNA loading into EVs. Binding of miR-1289 to the zipcode of its targets allows mRNAs to be loaded into EVs to reduce the cellular expression of these targets [69,70,71].

Overexpression of a desired gene in donor cells seems to prompt gene loading into EVs at both mRNA and protein level [72]. Significant therapeutic results were obtained using EVs derived from these cells against pancreatic cancer cells [73]. Recently, Yang et al. used an alternative method to produce therapeutic mRNA-containing EVs. In brief, using a cellular-nanoporation technique the authors transfected various source cells with plasmid DNAs and stimulated cells with an electrical stimulus to encourage the release of EVs loaded with the desired mRNAs. This method produced up to 50-fold more EVs and an over 103-fold increase in vesicular mRNA transcripts compared to previous strategies [74].

Active and specific RNA loading into EVs can also be performed using the Targeted and Modular EV Loading (TAMEL) platform (Figure 2). In this case, a plasmid encoding for the fusion protein composed of Lamp2b (an EV-enriched protein) and the RNA-binding domain (RBD) MS2 was transfected. This fusion protein was able to bind engineering RNAs-cargo (with RBD loops in the 3′ UTR) in the interior of EVs, enhancing the specific recognition. When using this method, it was observed that shorter RNAs were loaded into EVs with higher efficiency than longer molecules, confirming that loading capacity is inversely proportional to size [68].

In 2018, a novel parental cell-based strategy for loading mRNAs, known as EXOtic (Exosomal transfer into cells) was designed (Figure 3) and used to treat Parkinson’s disease-derived inflammation with encouraging results [75]. EXOtic aimed to achieve specific packaging/delivery of RNA and targeting. For specific RNA packaging, the RNA binding protein L7Ae was conjugated with CD63 and the RNA of interest was modified to allow L7Ae recognition. In order for EVs to reach only brain cells, rabies viral glycoprotein (RVG)-Lamp2b fusion protein was expressed in donor cells. Overexpression of the gap junction protein connexin 43 (Cx43) enhanced EV docking/uptake by receiving cells and further helped EV content to pass directly to receiving cells. Remarkably, subcutaneous implanting of engineered donor cells into a mouse resulted in reduced brain inflammation, confirming the promising potential of this method [75]. Another approach to achieve loading of miRNAs into EVs is the TAT–TAR protein–RNA interaction [76]. The authors fused a pre-miR-199a to a transactivating response (TAR) sequence and a transactivator of transcription (TAT) peptide of HIV-1 to Lamp2a (Lamp2a-TAT). When expressed in parental cells, miR-199a-TAR binds to Lamp2a-TAT and the interaction on the luminal C-terminal of Lamp2a leads to the effective loading of miR-199a into the lumen of EVs. RNA-binding modules seem to be an efficient method for loading RNA into EVs. For example, the RNA-binding protein HuR was fused to the C-terminal of CD9 in order to be localized in the exosomal lumen. Thus, HuR strongly binds to miR-155 and simultaneously they localize inside EVs [77].

### 2.2. EV Surface Functionalization

In 1986, Matsumura and Maeda described the “enhanced permeability and retention” (EPR) effect, which represented a passive delivery of carriers and/or therapeutics [78]. A major flaw of this effect was the lack of specificity that drove therapeutics towards off-target sites. EV surface modifications might overcome this limitation by displaying ligands onto the vesicle surface that recognize unique markers or abnormally expressed proteins on target cells, such as CD34 in AML blasts [79] and CD49 in prostate cancer cells [80]. Table 1 lists the most used ligands in therapeutic delivery against cancers. Based on this development, EVs were functionalized using a fusion protein composed of Lamp2b and the iRGD peptide to recognize αv integrins on the surface of breast cancer cells. Modified EVs were loaded with therapeutics and successfully reached the target cells [81]. However, an easier way to favor therapeutic delivery to cancer cells might be the use of cancer cell-derived EVs themselves, given their membrane composition comparable to that of donor cells [82] and high vesicle uptake [83]. However, rational reservations and unsafe implications suggest care in cancer cell-derived EVs therapeutic use. Chemistry methods were also applied in this field. Table 2 lists chemical reactions performed for ligand applications on vesicles, specifically with regard to cross-linkers. Polyethylene glycol (PEG) is the most commonly used cross-linker in therapeutic delivery [84,85,86]. CP05 is a CD63-specific peptide that acts as an anchor between vesicle membrane and ligand. It has been shown that is possible not only to isolate exosomes from human serum but also to modify the cargo (e.g., binding phosphorodiamidate morpholino oligomer (PMO)) for a specific therapeutic use [82]. TEG and C7 (2-aminobutyl-1-3-propanediol) were applied to attach molecules on the EV surface due to their affinity for phospholipid membrane [87,88]. An outdated physical method was employed to add antibodies to the EV surface through sonication. This technique allowed binding to the vesicle of a low percentage of antibodies (4–40%). Binding efficiency depends on the time and power of sonication and on the lipidic composition carried by EVs [89]. It was also recently reported that the presence of nucleases inside EVs might degrade therapeutic nucleic acids and impair their action [88]. Another possibility could be loading therapeutics onto the EV surface, depending on their modality of action (i.e., binding/activating receptors on the cell surface). Table 3 lists therapeutics added onto vesicle surface and successfully delivered to target cells. Haraszti et al. [88] loaded cholesterol-conjugated siRNA onto isolated EVs for Huntingtin mRNA silencing in primary neurons with promising results. Although the conjugation was not tested on tumor cells, its application in cancer treatment could be possible. Other membrane-conjugated siRNAs were tested on lung cancer cells [90] to inhibit the expression of CD45.

## 3. Engineering EV-Based Cancer Therapy

EV-based engineering strategies were developed for the treatment of both solid and hematologic cancers and hold big promises, as they might be more specific, more effective, and safer for patients compared to drug treatments. The main aim of EV-based gene cancer therapy is to transfer to the tumor site nucleic acid molecules, such as siRNA, miRNA, RNA, and DNA, in order to modify or manipulate the altered expression of target genes, reverting the aberrant cancer phenotype or conferring efficacy and safety to therapeutic agents. Table 4 lists the acid nucleic molecules used as therapeutic in cancer.

### 3.1. Solid Tumors

Most studies report that the use of miRNA and siRNA reduces the tumorigenicity of various types of solid cancers [113,114,115,116,117,118,119]. These molecules mediate RNA interference, inhibiting the translation of mRNA targets. Given their greater specificity [120], they can be considered the best replacement therapy for inhibitory drugs. Since angiogenesis is a key mechanism in the development of cancer, many research efforts have focused on the inhibition of genes involved in this process. One of the main target genes in angiogenesis is vascular endothelial growth factor (VEGF). miR-100/497 mediated downregulation of this gene leads to a reduction in the growth and spread of lung and breast cancer [121,122]. Specifically, the paracrine effects of miR-100 from MSC-derived EVs in breast cancer cells is indirectly involved in downregulation of VEGF through inhibition of mTOR and a consequent reduction in hypoxia-inducible factor (HIF) 1α [121]. Similarly, miR-497-loaded EVs repressed angiogenesis in lung cancer via direct interaction with VEGF-A and other factors, such as YAP1, HDGF, and CCNE1, in a 3D microfluidic device [122]. Inhibition of angiogenesis was also observed in the treatment of gastric cancer using siRNA against hepatocyte growth factor (HGF); in this case, the EV-siRNA system suppressed not only angiogenesis but also tumor growth, both in vitro and in vivo, through downregulation of HGF and subsequently of VEGF [123]. Furthermore, it was recently reported that EV-siRNA against VEGF was able to easily and effectively cross the blood–brain barrier in a zebrafish model of brain cancer. This study demonstrated the stability of EVs in circulation and their ability to act as couriers for the delivery of brain drugs, supporting a new strategy for cancer therapy [124]. A wide range of genes are involved in cancer spread, such as *COX-2*, known to be responsible for breast cancer pathogenesis, angiogenesis, and metastasis. Injecting EVs containing miR-379 reduced tumor growth in mice, revealing the effective action of miR-379 as a tumor suppressor [125]. Further studies reported that EV-delivered TRPP2 siRNA blocks cell migration and invasion of laryngeal cancer [126] and thatHSP27 siRNA tagged-EVs promote neuronal maturation and differentiation as well as a reduction in cell proliferation and viability in neuroblastoma cells [127]. In another interesting work, therapeutic biomimetic nanoparticles composed of cationic bovine serum albumin (CBSA)-conjugated siS100A4 and exosome membrane-coated nanoparticles, called CBSA/siS100A4@Exosome, was able to suppress postoperative breast cancer metastasis [128]. In addition, EVs derived from miR-148b-3p-overexpressing human umbilical cord MSCs and miR-134-enriched EVs reduced cellular migration and invasion, and enhanced drug sensitivity by inhibiting TRIM9, and targeting STAT5B, Bcl-2, and Hsp90, respectively [13,129]. Another important opportunity is to target oncogenic drivers. KRAS is one of the most representative and well-studied genes, found mutated in 30% of all human cancers. Several studies showed that EV-delivered mutated KRAS silencing efficiently reduces tumor proliferation and growth in lung and pancreatic cancer cells and in mouse tumor xenografts [130,131,132,133]. In a work on murine sarcoma, the use of a siRNA against transforming growth factor beta, delivered to the tumor via EVs, was shown to reduce cancer growth and repressed tumor progression [134]. Promising results using engineered EVs have been achieved in neuronal diseases not responsive to conventional drugs. In one of the first studies, miR-146b-delivered EVs inhibited glioma multiforme growth by targeting epidermal growth factor receptor (EGFR) mRNA in rats [135]. In addition, EVs derived from miR-199a-overexpressing MSCs inhibited glioma development by inducing apoptosis via downregulation of ankyrin repeat and pH domain 2 expression [136]. The most encouraging results of engineered EVs regard their ability to improve existing drug therapies by increasing sensitivity to conventional drugs to which patients acquire resistance. For instance, miR-199a and miR-9 released via EVs enhanced chemosensitivity in a combinatorial protocol with temozolomide, leading to a significant reduction in proliferation, migration, and invasion of glioma [136,137]. In breast cancer, the use of exosomal miR-567 proved to be an effective means of reversing resistance to trastuzumab by inhibiting ATG5 expression [138] and miR-134 increased cisplatin-induced apoptosis by enhancing sensitivity to anti-Hsp90 drugs [13]. In liver cancer, miR-122 was found to be remarkably effective in decreasing chemoresistance by downregulating three genes, ADAM10, IGF1R, and CCNG1, involved in tumorigenesis and drug sensitivity. The authors observed a decrease in tumor volume and weight, in vivo, exclusively through the use of miR-122 from adipose tissue-derived MSC EVs combined with sorafenib [139] for GRP78-siRNA EVs transfected into hepatocellular carcinoma cells, leading to a significant reduction in sorafenib resistance and tumor metastasis formation [140]. Finally, EV-released miR-128-3p increased chemosensitivity to oxaliplatin in colorectal cancer by negatively regulating expression of Bmi1 and MRP5 genes [141].

Besides delivering miRNAs and siRNAs, EVs can also carry mRNAs, long noncoding RNAs, and circular RNAs (circRNAs). These long noncoding RNAs are the subject of increasing interest because of their biological stability and function as gene expression regulatory elements when acting as a miRNA sponge. circ-0051443 delivered by EVs in hepatocarcinoma cells was reported to reduce tumor progression and, in vivo, the size and weight of xenograft tumors. In this case, circRNA acted as a sponge of miR-331-3p, inhibiting expression of the *BAK1* gene, an important cell death regulator [142]. EV circRNA was also found to play a role in chemoresistance. In gastric cancer cells, the long noncoding RNA HOXA transcript at the distal tip (HOTTIP) was able to promote resistance to cisplatin by regulating miR-218, which targets the HMGA1 gene, a tumor promoter in several cancers [143]. In colorectal cancer, hsa_circ_0005963 acted as a miR-122 sponge, causing overexpression of the PKM2 gene and consequently resistance to oxaliplatin [144]. One of the limitations associated with EV therapy concerns its production site. Most EVs are generated from primary MSCs or immortalized cells. Millions of cells are required to produce a sufficient quantity of EVs, severely hindering their potential therapeutic use. This limitation could be overcome by using EVs derived from human red blood cells (hRBC). Since RBCs constitute the majority of the body’s cells (about 84%) [145], a sufficiently high number of EVs can be produced while simultaneously abolishing immunological cytotoxicity, resulting from the use of primary cell-derived EVs. This approach was validated in a study by Usman et al., where hRBC-derived EVs proved to be an effective and safe method of gene therapy. As well as demonstrating that it is possible to target a specific miRNA, miR-125, by transferring an inhibitor via RBC-EVs in a particularly aggressive breast cancer line, the authors described a novel use of the CRISPR-Cas9 system in gene therapy. By transferring both Cas9 mRNA and RNA to the genetic target of interest, they obtained promising results in gene expression modulation [146].

### 3.2. Hematologic Tumors

Currently, in terms of gene therapy, the most cutting-edge therapeutic approach for hematological diseases is chimeric antigen receptor T cell therapy (CAR-T) [147]. Since most leukemia is caused by an imbalance of coding and noncoding genes [148], the development of opportune RNA interference therapies is required. One of the most studied targets in hematological diseases is c-Myc, and EVs targeting c-Myc by siRNA-induced apoptosis in lymphoma cells in vivo [149]. In addition, since inflammations are a key process in the development and progression of hematologic cancers, many research efforts are aimed at targeting monocytes and macrophages. One of the first studies in this field showed the effective transfer of a siRNA from plasma EVs of peripheral blood mononuclear cells into the target cells, causing selective gene silencing of MAPK-1 [150]. Dendritic-derived EV miR-155 and miR-146a can be effectively transferred between immune cells in vivo, and positively and negatively regulate the inflammatory process in mammals, respectively [151]. In a more recent study, the viability of precursor B cell acute lymphoblastic leukemia was reduced using engineered EVs carrying specific RNA oligonucleotides against Che-1/AATF polymerase [152]. However, one of the most interesting studies is that by Usman et al., where in addition to demonstrating the advantages of using EVs-derived from blood cells for gene therapy, the authors showed that induction of antisense miR-125 reduces the progression of acute myeloid leukemia (AML) in vivo. For this purpose, they loaded antisense oligonucleotide into 3 × 10^12^ EVs which subsequently were injected into tumor-bearing mice.

Notably, most of the articles in this review use a large amount of vesicles for in vivo experiments (ranging from 5 × 10^10^ to 10 × 10^12^ total EVs) without incurring toxicity, immunogenicity and changes in cytokine levels. Furthermore, EV purity evaluations were performed by different technologies (TEM, SEC, NanoSight) to ensure the therapeutic effects were due only to vesicles.

**Table 4 pharmaceutics-12-00980-t004:** List of acid nucleic molecules used as therapeutic in cancer, and related targets.

Therapeutic	Target(s)	Cancer Type	Outcome(s)	Reference
miR-100	VEGF, mTOR, HIF1α	Breast	reduction in the growth and spread	[121]
miR-497	VEGF, YAP1, HDGF, and CCNE1	Lung	angiogenesis repressed	[122]
siRNA	HGF	Gastric	angiogenesis and tumor growth suppressed	[123]
siRNA	VEGF	Brain	angiogenesis decreased	[124]
miR-379	COX-2	Breast	reduction in tumor growth	[125]
siRNA	TRPP2	Laryngeal	migration and invasion blocked	[126]
siRNA	HSP27	Neuroblastoma	neuronal maturation and differentiation promoted; reduction proliferation and viability of cancer	[127]
CBSA/siS100A4	S100A4	Breast	postoperative cancer metastasis suppressed	[128]
miR-148b-3p	TRIM59	Breast	proliferation, invasion, and migration inhibithed; apoptosis promoted	[129]
miR-134	STAT5B	Breast	cellular proliferation reduced; cisplatin-induced apoptosis enhanced	[13]
siRNA	TGF-β	Murine Sarcoma	cancer growth and tumor progression repressed	[134]
siRNA	KRAS	Lung and Pancreatic	tumor proliferation and growth reduced	[130,131,132,133]
miR-146b	*EGFR*	Glioma multiforme	Growth inhibited	[135]
miR-199a	AGAP2	Glioma	inducing apoptosis enhanced chemosensitivity for temozolomide	[136]
Anti-miRNA	miR-9	Glioma	proliferation, migration, and invasion reduced	[137]
miR-567	ATG5	Breast	reversing resistance to trastuzumab	[138]
miR-122	*ADAM10*, *IGF1R*, and *CCNG1*	Liver	decreasing chemoresistance; tumor volume and weight decrease	[139]
siRNA	GRP78	Hepatocellular carcinoma	reduction in sorafenib resistance and tumor metastasis formation	[140]
miR-128-3p	Bmi1, MRP5	Colorectal	increased chemosensitivity to oxaliplatin	[141]
circ-0051443	BAK1	Hepatocarcinoma	tumor progression and size and weight of tumor reduced	[142]
HOTTIP	miR-218	Gastric	promote resistance to cisplatin	[143]
hsa_circ_0005963	miR-122	Colorectal	promote resistance to oxaliplatin	[144]
siRNA	c-Myc	Lymphoma	Apoptosis induced	[149]
miR-155; miR-146a	SHIP1 and BACH1; IRAK1	BMDCs	regulate inflammation	[151]
RNA OLIGOS	Che-1/AATF	CLL	Viability reduced	[152]
125b-ASO	miR-125	AML	Progression reduced	[146]

## 4. Conclusions

Despite greater insights and rapid progress in the diagnosis and treatment of cancer, the efficacy of current treatment strategies is still limited. The synergism of gene therapy and EV communication has opened up new horizons in the treatment of malignancies. EV-based gene therapy represents a promising approach to delivering new precision medicine treatments for solid and hematological cancers. Crucially, EVs are highly stable, non-toxic, and non-immunogenic natural targeted delivery vehicles. Exploiting the properties of EVs in combination with gene therapy has already proved to be effective in efficiently targeting and delivering therapeutic agents in preclinical models. Further molecular and in vivo studies will be required to substantiate and validate current findings and improve EV-mediated gene therapy.

## Figures and Tables

**Figure 1 pharmaceutics-12-00980-f001:**
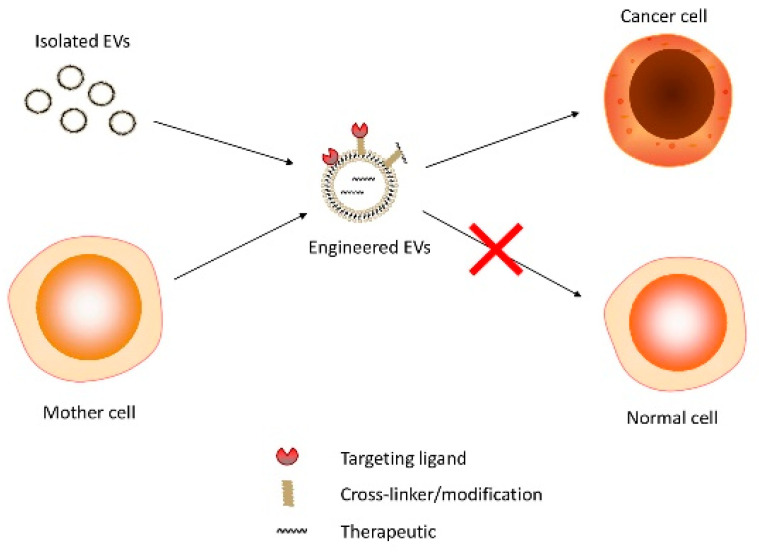
Therapeutic delivery: from source to target cell.

**Figure 2 pharmaceutics-12-00980-f002:**
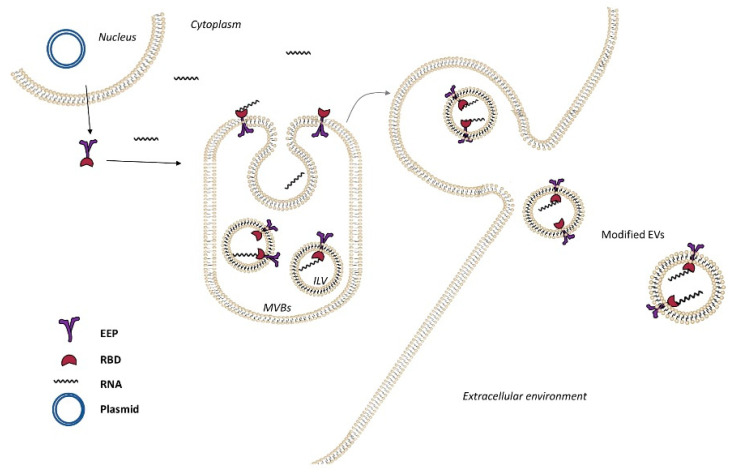
Active loading of cargo RNA into extracellular vesicles (EVs) via TAMEL. TAMEL EV-loading protein comprises an EV-enriched protein (EEP, violet) fused to an RNA-binding domain (RBD, red), which localizes to EVs. Actively loaded RNA contains a motif that binds to the RBD, resulting in enhanced loading into EVs.

**Figure 3 pharmaceutics-12-00980-f003:**
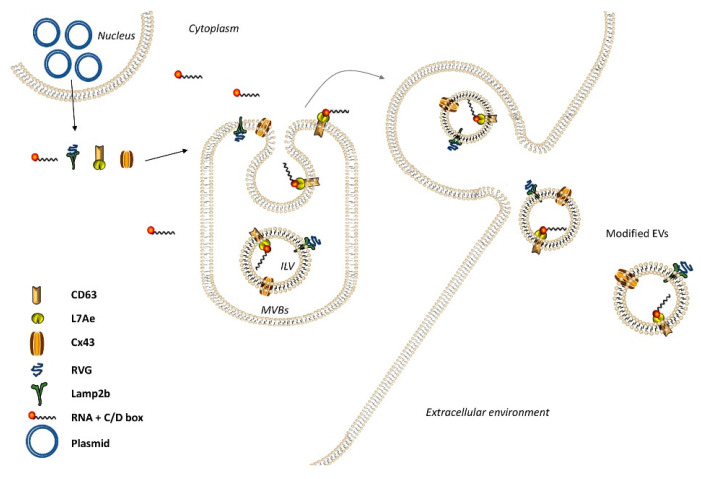
mRNA delivery by Exosomal transfer into cells (EXOtic). EV containing the RNA packaging device CD63-L7Ae (which can bind to the C/Dbox in the 3′ UTR of any RNA structure), targeting module (RVG-Lamp2b to target CHRNA7), cytosolic delivery helper (Cx43 S368A), and mRNA (e.g., nluc-C/Dbox) efficiently produced from EV donor cells. Engineered EVs will be delivered to target cells, expressing CHRNA7, with the help of Cx43.

**Table 1 pharmaceutics-12-00980-t001:** List of ligands and related targets in cancer therapy.

Molecule	Ligand(s)	Target(s)	Cancer Type(s)	Reference
Peptide	RGD	Integrins αvβ3, αvβ5, α5β1	Breast, glioblastoma, prostate, pancreas	[81,91,92,93]
Peptide	H2009.1	Integrin αvβ6	Lung, ovarian, oralcavity	[94]
Peptide	RVG	CHRNA7	Brain	[75]
Peptide	Bombesin	GRP receptor	Breast, lung, prostate	[95]
Peptide	SP94	-	Hepatocellular carcinoma	[96]
Peptide	GE11	EGFR	Breast, lung, hepatoma	[97,98,99]
Tripeptide	NGR	Aminopeptidase N	Blood vessels	[100]
Protein	Transferrin	Transferrin-receptor	Metastatic and drug-resistant cancer cells	[101]
Protein	EGF	EGFR	Breast	[97]
Ab	Anti-CD20	CD20	Burkitt’slymphoma	[102]
Ab	Anti-CD47	CD47	Pancreas	[103]
Ab	Anti-annexin A2	AnnexinA2	Breast, glioblastoma	[104]
Aptamer	AS1411	Nucleolin	Breast, non-small cell lung	[105]
Aptamer	Sgc8	PTK7 membrane protein	Leukemia	[106]
Aptamer	HeA2_1 HeA2_3	HER2 receptor	Breast	[107]
Monosaccharide	Galactose	Asialoglycoproteinreceptor	Liver	[108]
Glycosaminoglycan	Hyaluronic acid	CD44	Melanoma, colon, lung	[109]
Vitamin	Cobalamin	Transcobalaminreceptor	Lung, breast, pancreas	[110]
Vitamin	Folate	Folate receptor	Ovarian, breast, kidney, brain	[111]

**Table 2 pharmaceutics-12-00980-t002:** Chemical methods of extracellular vesicle (EV) surface cross-linking.

Method	Binding
Triazole linkage	Covalent
Disulfide linkage	Covalent
Thioether bond	Covalent
Amide bond	Covalent
Streptovidin-Biotin interaction	Non-covalent
π-π stacking interactions	Non-covalent

**Table 3 pharmaceutics-12-00980-t003:** Therapeutics conjugated to EV surface and their targets in cancers.

Molecule	Therapeutic	Target	Cancer	Reference
RNA	hsiRNA	Huntingtin	-	[88]
RNA	siRNA	HuR	Neuroblastoma	[112]
RNA	siRNA	CD45	Lung	[90]

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
