# Peer review of "Extracellular Vesicle-Based Nucleic Acid Delivery: Current Advances and Future Perspectives in Cancer Therapeutic Strategies"

_pharmaceutics, 2020, doi:10.3390/pharmaceutics12100980_

Round 1

Reviewer 1 Report

This manuscript by Massaro et al. reviews the developments in extracellular vesicle-mediated gene delivery for cancer therapy. The topic is interesting and the work is generally well presented. I would ask the reviewers to consider the following points before publication:

  1. Section 3 could be greatly improved by including a table summarizing the different types of nucleic acids included in EVs for each work described, together with the disease in which it was employed and perhaps the main outcome/conclusion.
  2. Throughout the manuscript, many words appear to be joined together when they should be separated (as examples, "MSCsinhibitedglioma", "generatedfrom"). This is probably a result of a change of format, but I would ask the authors to thoroughly revise the text to correct these typographical errors. 

Reviewer 2 Report

This is an enlightening review about EV-based nucleic acid delivery, with only some minor flaws.

Two general comments:

  1. Many examples are mentioned about the use of EVs for nucleic acid delivery, but quantitative description of the EV samples is lacking. For example, what are the typical EV and cargo concentrations in these studies, and is the purity of EV preps discussed? Are these quantities relevant and/or feasible for in vivo applications?
  2. The authors also mention in their review that EVs are mostly accumulated in the liver. This raises the question: how to use them for delivery to organs other than the liver? Can the authors discuss this issue in detail?

A minor comment: line 30: EVs cover a much larger length scale.

Reviewer 3 Report

The reviewer has no specific comments beside this Review is very well written and cover the current literature of the use of extracellular vesicles as a tool for therapeutic nucleic acid delivery.

Author Response

Response to Reviewer 3 Comments

The reviewer has no specific comments beside this Review is very well written and cover the current literature of the use of extracellular vesicles as a tool for therapeutic nucleic acid delivery.

We thank you for very careful review and kind comments.